# CH vs. HC—Promiscuous Metal Sponges in Antimicrobial Peptides and Metallophores

**DOI:** 10.3390/molecules28103985

**Published:** 2023-05-09

**Authors:** Kinga Garstka, Valentyn Dzyhovskyi, Joanna Wątły, Kamila Stokowa-Sołtys, Jolanta Świątek-Kozłowska, Henryk Kozłowski, Miquel Barceló-Oliver, Denise Bellotti, Magdalena Rowińska-Żyrek

**Affiliations:** 1Faculty of Chemistry, University of Wrocław, F. Joliot-Curie 14, 50-383 Wrocław, Poland; kinga.garstka2@uwr.edu.pl (K.G.); valentyn.dzyhovskyi@uwr.edu.pl (V.D.); joanna.watly2@uwr.edu.pl (J.W.); kamila.stokowa-soltys@uwr.edu.pl (K.S.-S.); henryk.kozlowski@uwr.edu.pl (H.K.); blldns@unife.it (D.B.); 2Faculty of Health Sciences, University of Opole, 68 Katowicka St., 45-060 Opole, Poland; jolanta.swiatekkozlowska@uni.opole.pl; 3Department of Chemistry, University of Balearic Islands, Cra. de Valldemossa, km 7.5., 07122 Palma de Mallorca, Spain; miquel.barcelo@uib.es; 4Department of Chemical, Pharmaceutical and Agricultural Sciences, University of Ferrara, I-44121 Ferrara, Italy

**Keywords:** bioinorganic chemistry, antimicrobial peptides, metallophores, thermodynamics, potentiometry, metal coordination

## Abstract

Histidine and cysteine residues, with their imidazole and thiol moieties that deprotonate at approximately physiological pH values, are primary binding sites for Zn(II), Ni(II) and Fe(II) ions and are thus ubiquitous both in peptidic metallophores and in antimicrobial peptides that may use nutritional immunity as a way to limit pathogenicity during infection. We focus on metal complex solution equilibria of model sequences encompassing Cys–His and His–Cys motifs, showing that the position of histidine and cysteine residues in the sequence has a crucial impact on its coordination properties. CH and HC motifs occur as many as 411 times in the antimicrobial peptide database, while similar CC and HH regions are found 348 and 94 times, respectively. Complex stabilities increase in the series Fe(II) < Ni(II) < Zn(II), with Zn(II) complexes dominating at physiological pH, and Ni(II) ones—above pH 9. The stabilities of Zn(II) complexes with Ac-ACHA-NH_2_ and Ac-AHCA-NH_2_ are comparable, and a similar tendency is observed for Fe(II), while in the case of Ni(II), the order of Cys and His does matter—complexes in which the metal is anchored on the third Cys (Ac-AHCA-NH_2_) are thermodynamically stronger than those where Cys is in position two (Ac-ACHA-NH_2_) at basic pH, at which point amides start to take part in the binding. Cysteine residues are much better Zn(II)-anchoring sites than histidines; Zn(II) clearly prefers the Cys–Cys type of ligands to Cys–His and His–Cys ones. In the case of His- and Cys-containing peptides, non-binding residues may have an impact on the stability of Ni(II) complexes, most likely protecting the central Ni(II) atom from interacting with solvent molecules.

## 1. Introduction

Cys–His and His–Cys motifs are quite abundant in antimicrobial peptides (AMPs), being present in 411 of such compounds (168 Cys–His and 243 His–Cys motifs, respectively). For the sake of comparison—348 Cys–Cys motifs and only 94 His–His motifs can be found in the AMP database [1].

Such peptide motifs are very tempting metal binding sites for metal ions such as Zn(II), Fe(II) or Ni(II), which play pivotal roles in bacterial and fungal cells. They are involved, for example in catalysis or structure functions in proteins, whose side chains or the terminal groups are able to bind metal ions [2,3]. The metal binding of the amino acids is selective and depends on the hard/soft character of the metal ion [2,4]. The most important metal anchoring sites are the side chains of histidine (imidazole) or cysteine (thiolate) residues and the carboxyl groups of glutamic and aspartic acids [5]. Thus, proteins that contain regions rich in histidine and cysteine residues are an interesting case of ligands due to a high affinity to bind Zn(II), Ni(II) and Fe(II) [6,7,8,9] and are chosen both by nature and by science as good metal chelators. Knowledge about the coordination mode, structure and function of specific binding sites in biomimetic artificial ligands with SH/NH groups and their metal complexes biomolecules provides us knowledge that can be transferred to a new generation of materials and applications, such as catalysis [10], bioassays [11] or general peptide-governed metal sorption [12,13]. It is important to note that often, seemingly very similar coordination motifs (as also in the case of this work) do not necessarily lead to the same coordination and functional outcomes, which are important factors to consider in the design of novel biomimetics, catalysts and materials [14].

Due to its structure and acid-base properties, histidine residue shows versatility in molecular interactions [15]. The functional group of the histidine residue is the aromatic imidazole ring (with a pK_a_ slightly above 6 [16]), which contains two nitrogen atoms [17]. At physiological pH (about 7.4), the imidazole group of histidine is partially deprotonated—observed in both acidic and basic forms. In the case of the acidic imidazolium cation, the positive charge is evenly distributed between the two nitrogen atoms. In response to a change in pH values, each of them can release a proton and form equivalent tautomer forms: N1-H or N3-H, capable of binding metal ions [15].

Depending on the location and amount of histidine residues in the chain of proteins and peptides, we observe different coordination modes of metal ions [15,18,19,20,21]. Peptides that contain a histidine residue in the first position can form complexes with one of the nitrogen atoms of the histidine imidazole ring and the nitrogen atom of the N-terminal amino group [22]. This coordination mode is called the histamine-like model and its characteristic feature is the formation of a five-membered ring [23]. The histamine coordination mode for Zn(II) is maintained in the entire pH range, and the zinc coordination sphere can be filled by oxygen, nitrogen and sulfur atoms derived from side groups of other amino acids or water molecules.

In the case of Cu(II) and Ni(II) (and, theoretically, also Fe(II)), which are capable of deprotonating the amide bond at basic pH, the metal coordination sphere may include amide nitrogen atoms, resulting in stable, six-membered rings with a {NH_2_, 3N^−^} mode [19,22,24,25]. When the histidine residue is in the second position, it is possible to form stable five- or six-membered rings that stabilize the coordination of Ni(II) and Cu(II). The coordination sphere includes two nitrogen atoms derived from the histidine residue (one from the imidazole of the histidine residue and one amide nitrogen atom), as well as the nitrogen atom of the N-terminal amino group and the oxygen atom derived from the carboxyl group present in sequence or water molecule [19]. If the histidine residue is on the third position in the sequence, such a motif is called the ATCUN (amino terminal Cu(II) and Ni(II) binding site) motif or albumin-like coordination site [26,27]. This binding site is particularly attractive for Cu(II) and Ni(II). The coordination sphere includes an imidazole nitrogen atom, two amide nitrogen atoms and an amine N-terminal nitrogen atom, which results in the formation of three chelate rings: one six-membered ring and two five-membered ones.

The position and number of the histidine residues in peptides are very important and determine the coordination mode of metal ions. In addition to the previously mentioned coordination modes, it is also worth mentioning that at any place in the chain, there may be a specific sequence of amino acids—HEXXH—with a high affinity for Zn(II) [28]. Its presence has been found, among others, in metalloproteinases dependent on zinc ions [29]. In this case, the coordination sphere includes two imidazole nitrogen atoms, and the coordination sphere may be filled by oxygen atoms from the Glu carboxyl group or by water molecules [30]. Moreover, usually, the more histidine residues a peptide contains, the higher the thermodynamic stability of its imidazole-coordinated complexes with metal ions [24,31,32].

Cysteine is an exceptional proteinogenic amino acid. It contains a reactive sulfhydryl group [33], and two cysteines may form a disulfide bond (cystine) [34]. The disulfide links between various residues present in different parts of the protein or between separated polypeptide chains induce the folding of proteins [33] and play a structural role in proteins. Due to the rigid planar nature of the peptide bond, two cysteine residues present next to each other or separated by only one residue are unable to interact. The separation of two cysteine residues by two amino acid residues causes their closest proximity in the α-helix and β-turn. This motif, C-(X)_2_-C, is commonly present in many proteins such as oxidoreductases (X may stand for any amino acid) [35]. In the case of peptides that contain both cysteine and histidine residues, such as Ac-HAAC-NH_2_ or Ac-CGAH-NH_2_ [5], the C-terminal donor group is the primary anchor group, and at higher pH values, the N-terminal group is replaced by a third amide nitrogen atom [36].

Zinc, after iron, is the second most important trace element in the human organism; [37] it is necessary for growth and cell division, not only in the human body, but also in all other forms of life. In contrast to other trace elements (such as copper or iron), Zn(II) does not change its oxidation state in solution (due to its electron configuration, [Ar]3d^10^, it does not show d-d transitions and, therefore, is not visible in absorption spectroscopy) [38]. Zn(II) also plays a significant structural role [28], for example, in proteins with zinc finger motifs [39]. The geometry of zinc complexes may be different, but the most common one is tetrahedral [40]. Zn(II)’s most preferred donor atoms are nitrogen, sulfur and oxygen present in the side chains of histidine, cysteine, glutamic or aspartic acid residues, respectively. The coordination sphere can be filled with water molecules.

Another biologically relevant metal ion is nickel(II). It is essential for many eubacteria, archaebacteria, fungi and plants because it has important functions in diverse metabolic processes (metabolism and virulence) [41]. Ni(II) plays a catalytic role in the cofactors of nine enzymes [42]. Humans do not have any Ni-dependent enzymes; however, this metal is present in negligible amounts in human bodies, most likely since it is necessary for commensal bacteria [43]. Large doses or prolonged contact with nickel could be genotoxic, haematotoxic, teratogenic, cancerogenic and immunotoxic [44]. From a chemical point of view, Ni(II) and Fe(II) are very interesting, because they could theoretically compete with Zn(II) for its binding sites.

Iron is an essential trace metal for most living organisms. It serves as a prosthetic group for proteins involved in central cellular processes, such as respiration, oxygen transport and DNA synthesis [45]. Non-heme iron is bound by a large number of enzymes found in the cytosol. There are two classes of these proteins, those containing iron-sulfur clusters and those without iron-sulfur clusters. In the case of non-sulfur cluster proteins, Fe(II) is coordinated by six donor atoms forming octahedral complexes or five donor atoms which form a trigonal bipyramidal geometry. For complexes with a coordination number equal to five, the coordination frequently involves open coordination sites for the simultaneous binding of substrates. Most often, nitrogen or oxygen donor atoms (e.g., histidyl residues, aspartic or glutamic acid residues) are involved in Fe(II) binding [46].

The results of our recent work [47], in which potentiometric and spectroscopic techniques were used to study the interactions of Ac-PNCHTHEGGQLHCT, the C-terminal part of the Aspf2 protein, with Zn(II) and Ni(II), encouraged us to pursue further studies on the topic of peptides with cysteine and histidine residues in different positions in the model peptides. In this work, we report results obtained in systematic studies of Zn(II), Ni(II) and Fe(II) complexes of two model peptides containing histidine, cysteine and alanine residues in their sequences. We checked the thermodynamic parameters of Ac-AHCA-NH_2_ and Ac-ACHA-NH_2_ peptides. The protonation constants and the stability constants of formed complexes were determined using potentiometry, and the obtained results allowed us to identify the species that are present in the solutions at different pH values. Independent spectroscopic methods (UV-Vis spectroscopy and CD spectroscopy) allowed us to determine precise Ni(II) binding sites. In the scope of this paper, we try to answer three most important questions. (i) His or Cys—does it matter for the stability of their complexes which one goes first? (ii) Do His–Cys and Cys–His ligands have a metal preference? The comparison with some previously studied peptides which contain histidine and/or cysteine residues in different positions will allow us to answer the last and equally important question: (iii) Do non-binding residues affect complex stability? These data may provide an overview of the metal binding ability of -HC- and -CH- motifs and may help to better understand how Zn(II) is able to compete with other metal ions for binding sites in proteins.

## 2. Results and Discussion

The structural and thermodynamic properties of Zn(II), Ni(II) and Fe(II) complexes with Ac-AHCA-NH_2_ and Ac-ACHA-NH_2_ were studied using potentiometry, UV-Vis and CD spectroscopy. Potentiometric titrations allowed us to determine the ligand protonation constants, the stability constants of the formed complexes and the pH-dependent distribution diagrams. The combined UV-Vis and CD spectroscopic results clarified the binding mode of Ni(II) and the geometry of the species present in solution. Consequently, the combination of all these methods allowed us to explain coordination geometries and perform a detailed thermodynamic analysis of Ac-AHCA-NH_2_ and Ac-ACHA-NH_2_ peptides.

### 2.1. Protonation Constants of the Ac-AHCA-NH_2_ and Ac-ACHA-NH_2_

Both peptides are protected in the N-terminus through acetylation and in the C-terminus through amidation and contain two possible sites of protonation—a His and a Cys residue; their pK_a_ values are reported in Table 1. The first protonation constant for Ac-AHCA-NH_2_ corresponds to the imidazole nitrogen atom of histidine residue (6.42) and the second one, with the value 8.72, to the thiolate of the cysteine residue. The Ac-ACHA-NH_2_ peptide also behaves as an H_2_L acid with the deprotonating groups corresponding to the His and Cys residues with pK_a_ values: 6.42 and 8.76, respectively (Table 1).

### 2.2. Metal Complexes of the Ac-AHCA-NH_2_ Peptide

In the case of the Zn(II)-Ac-AHCA-NH_2_ complex, the first detected species, with a maximum at pH 7, is ZnL (Table 1 and Appendix A). Most probably, in this form, the Zn(II) is coordinated to one imidazole nitrogen and one Cys thiol. The remaining coordination positions are likely occupied by water molecules. The next observed species, ZnH_−1_L (maximum at pH 8.90), has a {N_im_, S^−^, OH^−^} binding mode, and it forms with pK_a_ = 7.84, corresponding to the deprotonation of a coordinated water molecule. Above pH 10.0, ZnH_−2_L dominates in solution. This case is most likely related to the deprotonation of the second water molecule (pK_a_ = 9.98), and in this form, the metal ion coordination sphere includes the imidazole, the sulfur and two oxygen atoms from the water molecules (Figure 1A).

The first Ni(II) complex observed at low pH, NiL reaches its maximum at pH 7.5 (Table 1 and Appendix A). Most probably, in this complex, Ni(II) is bound to the His imidazole and Cys sulfur. CD spectra confirm this finding with pronounced bands around 270 nm and 350 nm, typical for N_im_ → Ni(II) [48] and S^−^ → Ni(II) [49] charge transfer transitions, respectively (Appendix A). The next deprotonation leads to the formation of NiH_−1_L species, in which an amide nitrogen binds to Ni(II). The loss of the next proton leads to the NiH_−2_L form and is most likely related to the deprotonation and binding of a second amide. The last observed form, NiH_−3_L, most likely corresponds to the deprotonation of a water molecule bound to the central Ni(II) ion. In this form, one cysteine thiolate, one histidine imidazole and two amide nitrogens already participate in binding, forming a square planar nickel complex (Figure 1B). The hypothesis is supported also by the UV-Vis spectra (Appendix A), where no changes in the wavelength of maximum absorption are observed above pH 8.5.

The coordination of Fe(II) ions begins at a pH of about 6, (Table 1, Appendix A). The first species observed at this pH, FeL, is related to the simultaneous deprotonation of two functional groups (His imidazole and Cys thiol). As Fe(II) ions have a preference for both nitrogen and sulfur donor atoms [50], both of these amino acid residues are most likely involved in coordination. As pH values increase, another complex, FeH_−2_L, is formed due to the simultaneous loss of two protons. Most likely, it is related to the deprotonation and participation in the binding of the metal ion of the amide nitrogen of the peptide backbone [50,51] (Figure 1C), although it is also possible that at this point, two water molecules bound to the central iron atom deprotonate; advanced studies on the possibility of involving amides in binding are forthcoming. In the last deprotonation step, an FeH_−3_L species is formed and, as in the case of the Ni(II) complex, the deprotonation of a water molecule bound to the metal ion possibly takes place.

**Figure 1 molecules-28-03985-f001:**
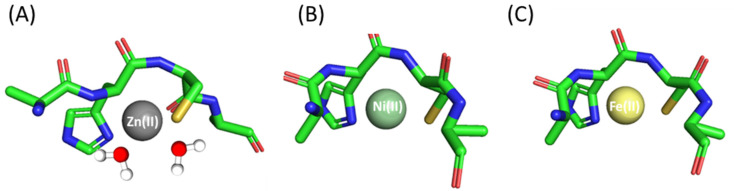
Proposed binding mode for the (**A**) Zn(II), (**B**) Ni(II) and (**C**) Fe(II) complexes with Ac-AHCA-NH_2_—a schematic model. The figure was generated using PyMOL [52].

### 2.3. Metal Complexes of the Ac-ACHA-NH_2_ Peptide

Ac-ACHA-NH_2_ coordination to Zn(II) results in three different species in the studied pH range: ZnL, ZnH_−1_L and ZnH_−2_L. ZnL is the first detected species, where most probably histidine and cysteine residues are already bound to Zn(II). This form has a maximum of around pH 7.0 (Table 1 and Appendix A). The second one, ZnH_−1_L, which dominates in solution in the pH range 8.0–10.0, is most likely related to the deprotonation of the first water molecule bound to the central Zn(II). Most probably, in the ZnH_−2_L complex, a subsequent coordinated water molecule deprotonates (Figure 2A).

In the studied pH range (2.5–10.5) and at a metal:ligand molar ratio of 1:1, we observed four different species of Ni(II) complex (Table 1 and Appendix A). The first species is a bis-complex with two ligand molecules bound to the metal (NiL_2_). This species occurs at pH 6.0 and achieves maximum concentration at around pH 8.0. Spectroscopic data obtained in this pH range, in particular the shoulders in the spectra at around 500 and 410 nm, suggest that in NiL_2_, Ac-ACHA-NH_2_ binds Ni(II) via four donor atoms in a {2S^−^, 2N_im_} binding mode. This is consistent with previously studied Ni(II) complexes with Cys- and His-containing peptides [53]. CD spectra also confirm this finding with pronounced d-d bands at 305 nm, which can be assigned to S^−^→Ni(II) charge transfer transitions [49]. In the NiH_−1_L form, apart from Cys and His side chains, an amide also takes part in metal binding, as confirmed by the presence of negative and positive bands at 550 and 480 nm in the CD spectra (Appendix A) [53,54,55]. Above pH 8.5, the square planar nickel complex reorganizes; the next two forms, NiH_−2_L and NiH_−3_L, most likely involve additional amide nitrogen atoms and a water molecule in binding, which can be confirmed via d-d transitions in the UV-Vis spectra at around 430 nm (Appendix A).

Fe(II)- Ac-ACHA-NH_2_ behaves similarly to the complex with a Ac-AHCA-NH_2_ ligand; only slight differences in logβ values were observed. Thus, the coordination process begins at pH 6 with the formation of an FeL species that reaches its maximum of formation at a pH of around 8.5, being also the only species present at physiological pH (pH 7.4). The Fe(II) ion is probably bound to sulfur and nitrogen donor atoms of cysteine and histidine residues, respectively, and the rest of the coordination sites are filled with either nitrogen amide or water molecules (Figure 2C).

To compare the strength of Zn(II), Ni(II) and Fe(II) binding to Ac-AHCA-NH_2_, Ac-ACHA-NH_2_ and other peptides described in the literature, we used their calculated stability constants to prepare so-called competition plots, which illustrate the comparison of the complexes’ stability. Such plots show a theoretical situation in which equimolar amounts of all reagents, metals and peptides are mixed. In the whole pH range, the Ac-ACHA-NH_2_ and Ac-AHCA-NH_2_ bind Zn(II) with comparable affinity (Figure 3A). In both cases, the coordination of Zn(II) involves one histidine imidazole and one cysteine thiolate, and the coordination sphere is filled with two water molecules. The results suggest that the location of the histidine and cysteine residues, which are next to each other, does not have a significant effect on the stability of the Zn(II) complexes—Zn(II) is equally ready to bind to the HC motif as it is to bind to the CH one.

Nickel’s preferences in binding Ac-ACHA-NH_2_ and Ac-AHCA-NH_2_ are substantially different (Figure 3B). Ni(II) binds to both ligands with comparable affinity up to pH 8, while at a higher pH, at which amides take part in binding, Ni(II) coordination to Ac-AHCA-NH_2_ is strongly preferred, suggesting that the {S^−^, 2N^−^} binding mode is more stable than the {N_im_, 2N^−^} one.

In the case of Fe(II) complexes, the trend is very similar to that of Zn(II) complexes; Fe(II) seems to bind to both Ac-ACHA-NH_2_ and Ac-AHCA-NH_2_ with comparable affinity (Figure 3C).

Which is the preferred Zn(II) binding site—CH/HC or CC? Figure 4A gives us a straightforward answer—the ‘CC’ motif clearly binds Zn(II) with higher affinity, at least in the compared sequences: Ac-CC-NH_2_, Ac-GSCCHTGNHD-NH_2_, Ac-EEGCCHGHHE-NH_2_ and Ac-CCSTSDSHHQ-NH_2_ [56].

Thiolates are clearly the most likely binding sites for Zn(II)—our HC/CH ligands (Ac-AHCA-NH_2_ and Ac-ACHA-NH_2_) bind this metal much more effectively than peptides which contain two His imidazoles (Ac-GHEITHG-NH_2_, Ac-GHTIEHG-NH_2_, Figure 4B), showing that the Zn(II)-N_im_, S^−^ complex turns out to be thermodynamically stronger than Zn(II)-2N_im_—in our hypothetical situation, at physiological pH (pH 7.4), nearly 50% of Zn(II) is bound to Ac-ACHA-NH_2_, nearly 40% of Zn(II) is bound to Ac-AHCA-NH_2_ and only around 15% is bound to Ac-GHEITHG-NH_2_ and Ac-GHTIEHG-NH_2_ [57].

Nickel’s preferences for CH- and HC-type ligands differ somewhat from those of Zn(II). Ni(II) forms very stable square-planar complexes with the CC motif, binding to two thiolates and the amide in between. Ni(II)-CC complexes are, as Zn(II) complexes, more stable than CH and HC ones, but for different reasons—in the case of Zn(II), the affinity for sulfur seems to be the reason for the high affinity, while in the case of Ni(II), the planar geometry and additional stabilization by the amide seem to play the major role (the competition plot between Ni(II), Ac-AHCA-NH_2_, Ac-ACHA-NH_2_, Ac-CC-NH_2_, Ac-GSCCHTGNHD-NH_2_, Ac-EEGCCHGHHE-NH_2_ and Ac-CCSTSDSHHQ-NH_2_ [56] is shown on Figure 5A).

What is the impact of the non-binding residues on the Ni–CH complex stabilities with a {N_im_, N^−^, S^−^} binding mode? As hypothesized, the non-binding residues are able to protect the complex from hydrolysis, most likely forming a network of hydrogen bonds that protects the central Ni(II) ion. In our hypothetical situation in Figure 5B, Ni(II) complexes with Ac-AHCA-NH_2_ and Ac-ACHA-NH_2_ are substantially weaker than those with Ac-GGKPDLRPCHP-NH_2_ and Ac-PCHYIPRPKPR-NH_2_ [58].

## 3. Experimental Section

### 3.1. Materials

The peptides (Ac-AHCA-NH_2_ and Ac-ACHA-NH_2_,) were purchased from Karebay Biochem (certified purity: 98%) and were used as received. Zn(II), Ni(II) perchlorate and ammonium iron(II) sulfate (Mohr’s salt) were extra pure products (Sigma-Aldrich, Poznań, Poland). The carbonate-free stock solution of 0.1 M KOH was purchased from Sigma-Aldrich and then potentiometrically standardized with potassium hydrogen phthalate.

### 3.2. Potentiometric Measurements

The stability constants for proton Zn(II) and Ni(II) complexes with ligands Ac-AHCA-NH_2_ and Ac-ACHA-NH_2_ were calculated from titration curves carried out over the pH range of 2–11 at 298 K and ionic strength 0.1 M NaClO_4_. The total volume of the solution used was 3.0 cm^3^. The potentiometric titrations were performed using a Dosimat 800 Metrohm Titrator connected to a Methrom 905 pH-meter and a Mettler Toledo pH inLab Science electrode. The thermostabilized glass cell was equipped with a magnetic stirring system, a microburet delivery tube and an inlet-outlet tube for argon. Solutions were titrated with 0.1 M carbonate-free NaOH. The electrodes were calibrated daily for hydrogen ion concentration through titrating HClO_4_ with NaOH using a total volume of 3.0 cm^3^. The purities and the exact concentrations of the ligand solutions were determined using the Gran method. The ligand concentration was 0.5 mM. The Zn(II), Ni(II) and Fe(II) to ligand ratio was 1:1. The HYPERQUAD 2008 program was used for the stability constant calculations. The standard deviations were computed using HYPERQUAD 2008 and referenced to random errors only. The constants for hydrolytic Zn(II) and Ni(II) species were used in these calculations. The speciation and competition diagrams were computed using the HYSS program [59].

In the case of measurements of Fe(II) complexes, all the experiments were carried out in degassed solvents. A freshly prepared Fe(II) ion stock solution was employed for each measurement. The argon with which the measuring cell is purged prevents both carbonate formation and Fe(II) oxidation.

### 3.3. Spectroscopic Studies

The absorption spectra were recorded on a Jasco-730 spectrophotometer, in the range 200–800 nm, using a quartz cuvette with an optical path of 1 cm. Circular dichroism spectra were recorded on a Jasco J-1500 CD spectrometer in the 200–800 nm range, using a quartz cuvette with an optical path of 1 cm in the visible and near-UV range. The concentration of sample solutions used for spectroscopic studies was similar to those employed in the potentiometric experiment. The metal:ligand ratio was 1:1. All spectroscopic measurements were recorded in the pH range 2.5–10.5. The pH of the samples was adjusted with appropriate amounts of concentrated HClO_4_ and KOH solutions. The UV-Vis and CD spectroscopy parameters were calculated from the spectra obtained at the pH values corresponding to the maximum concentration of each particular species, based on distribution diagrams. OrginPro 2016 was used to process and visualize the obtained spectra.

## 4. Conclusions

Zn(II), Ni(II) and Fe(II) are crucial for proper cell development. Therefore, it is important to understand which type of binding sites and which type of precise sequences bind them with the highest (or with ‘the most appropriate’) affinity. For this reason, we wanted to answer three questions: (i) His or Cys—does it matter (for metal complex stability) which one goes first? (ii) Do His–Cys and Cys–His ligands have a metal preference? (iii) Do non-binding residues have a large effect on complex stability?

The results of this work allowed us to come to the following conclusions: (i) The position of His and Cys residues affects the stability of the complexes with Ni(II), but not those with Zn(II) and Fe(II). Ni(II) will more likely bind to Ac-AHCA-NH_2_ than to Ac-ACHA-NH_2_ above pH 8.0, when amide nitrogen atoms begin to take part in coordination (Figure 6A,B). (ii) His–Cys and Cys–His ligands do have a strong metal preference; up to pH 8.5, Zn(II) complexes are clearly the strongest ones. At alkaline pH (above 9.0), in which amides start to bind Ni(II), the square planar Ni(II) species become strongest. Fe(II) complexes are the weakest ones, thus obeying the Irving–Williams series (Figure 6A,B). (iii) Non-binding residues have a clear enhancing effect on the stability of Ni(II) complexes, most likely through shielding the central Ni(II) ion from surrounding solvent molecules.

## Figures and Tables

**Figure 2 molecules-28-03985-f002:**
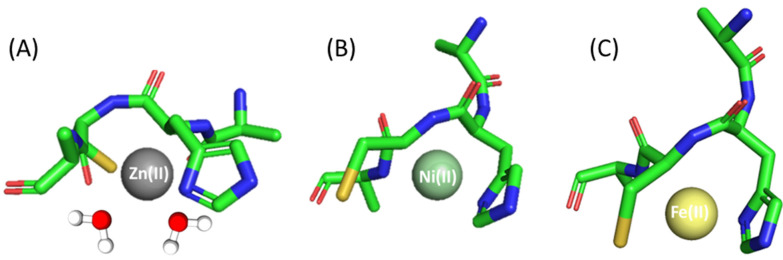
Proposed binding mode for the (**A**) Zn(II), (**B**) Ni(II) and (**C**) Fe(II) complexes with Ac-ACHA-NH_2_—a schematic model. The figure was generated using PyMOL [52].

**Figure 3 molecules-28-03985-f003:**
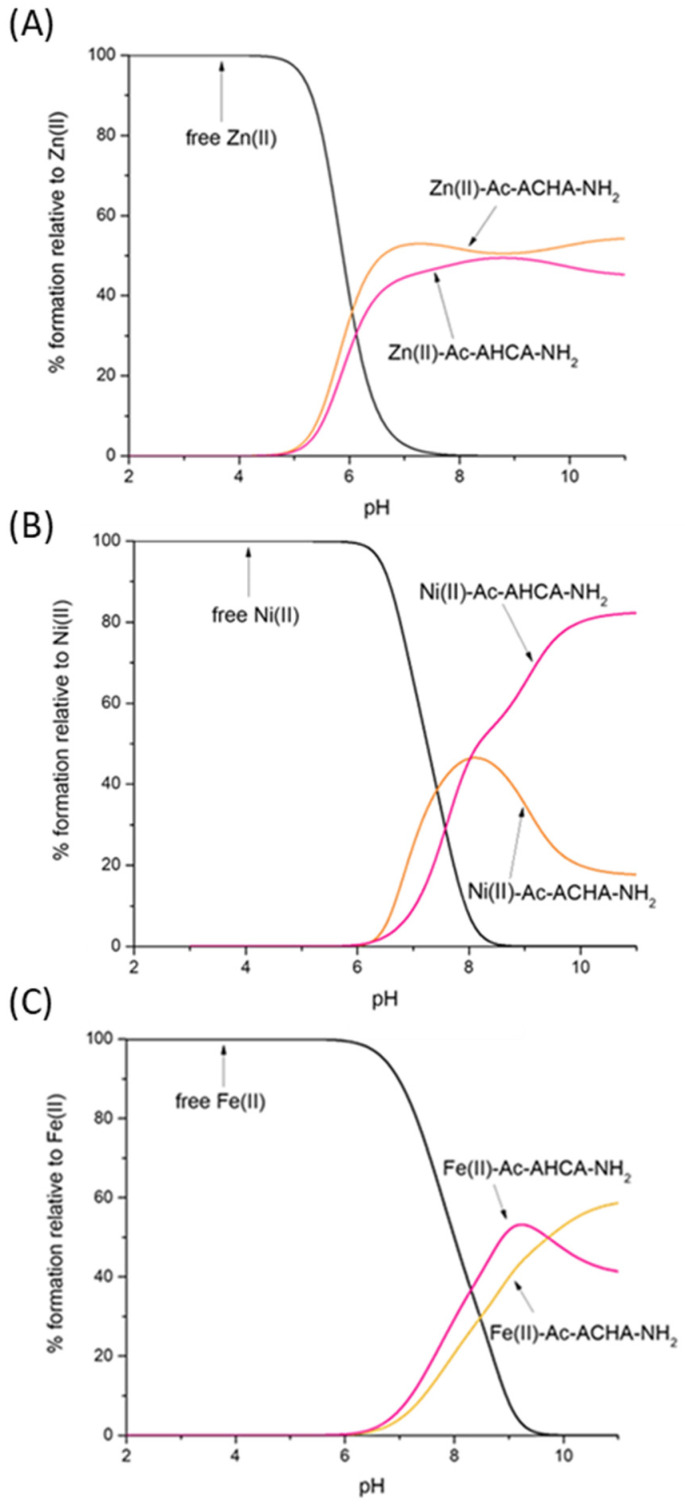
Competition plot between (**A**) the Ac-AHCA-NH_2_, Ac-ACHA-NH_2_ peptides and Zn(II); (**B**) the Ac-AHCA-NH_2_, Ac-ACHA-NH_2_ peptides and Ni(II), and (**C**) the Ac-AHCA-NH_2_, Ac-ACHA-NH_2_ peptides and Fe(II). The plot describes complex formation at different pH values in a hypothetical situation in which equimolar amounts of the three reagents are mixed. Calculations are based on constants from Table 1.

**Figure 4 molecules-28-03985-f004:**
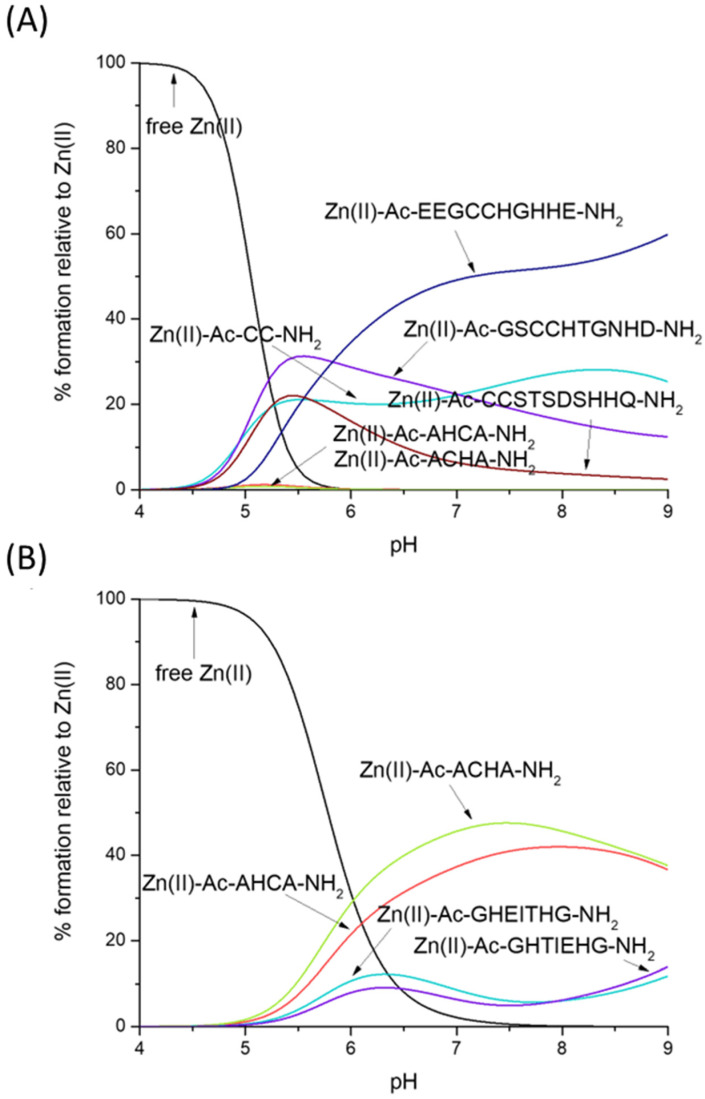
Competition plot between the Zn(II), Ac-AHCA-NH_2_, Ac-ACHA-NH_2_ peptides and (**A**) Ac-CC-NH_2_, Ac-GSCCHTGNHD-NH_2_, Ac-EEGCCHGHHE-NH_2_, Ac-CCSTSDSHHQ-NH_2_ [56] vs. (**B**) Ac-GHEITHG-NH_2_, Ac-GHTIEHG-NH_2_ [57]. The plot describes complex formation at different pH values in a hypothetical situation in which equimolar amounts of all reagents are mixed.

**Figure 5 molecules-28-03985-f005:**
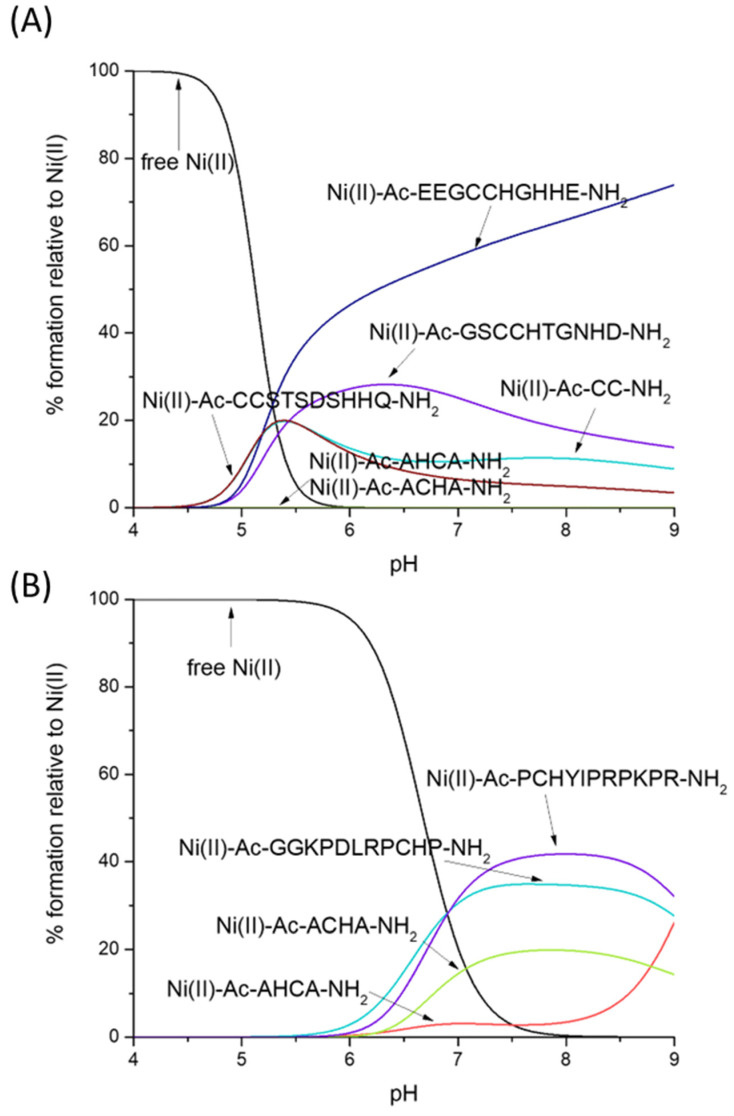
Competition plot between the Ni(II), Ac-AHCA-NH_2_, Ac-ACHA-NH_2_ peptides and (**A**) Ac-CC-NH_2_, Ac-GSCCHTGNHD-NH_2_, Ac-EEGCCHGHHE-NH_2_, Ac-CCSTSDSHHQ-NH_2_ [56] vs. (**B**) GGKPDLRPCHP-NH_2_, PCHYIPRPKPR-NH_2_, PCHPPCHYIPR-NH_2_ [58]. The plot describes complex formation at different pH values in a hypothetical situation in which equimolar amounts of all reagents are mixed.

**Figure 6 molecules-28-03985-f006:**
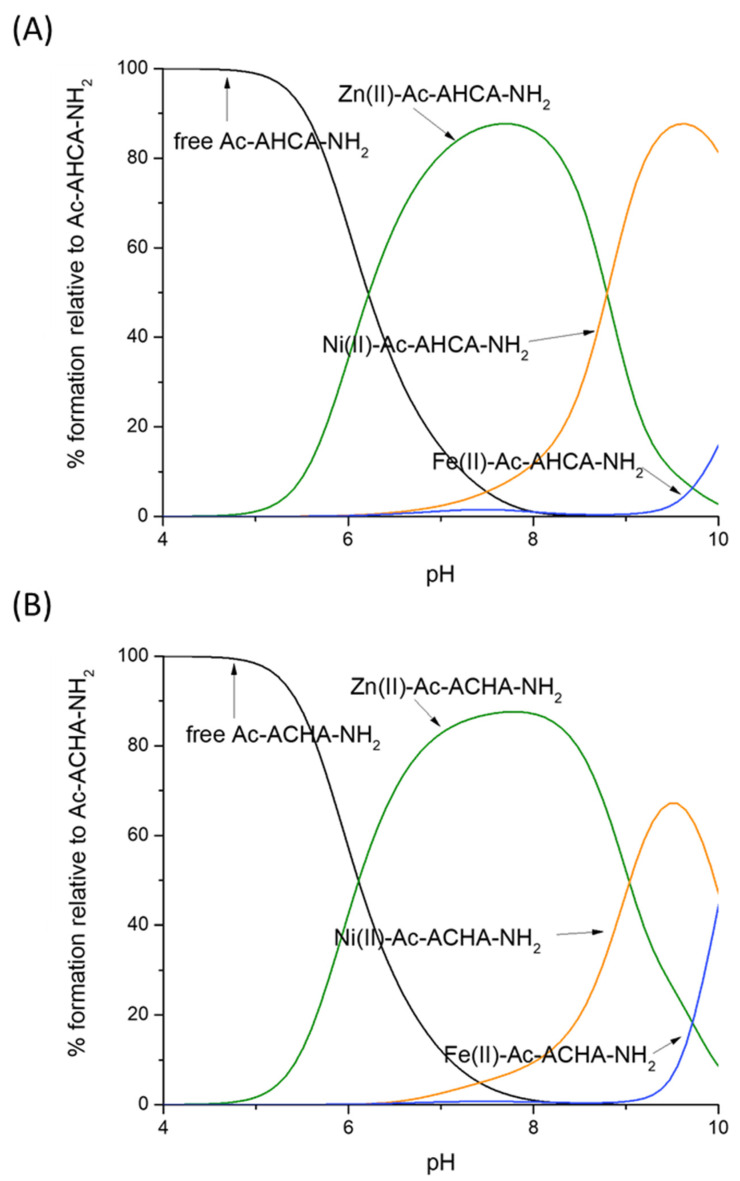
Competition plot between (**A**) the Ac-AHCA-NH_2_ peptide, Zn(II), Ni(II) and Fe(II) vs. (**B**) the Ac-ACHA-NH_2_ peptide, Zn(II), Ni(II) and Fe(II). The plot describes complex formation at different pH values in a hypothetical situation in which equimolar amounts of the three reagents are mixed. Calculations are based on constants from Table 1.

**Table 1 molecules-28-03985-t001:** Potentiometric data for the proton, Zn(II), Ni(II), Fe(II) complexes of the Ac-AHCA-NH_2_ and Ac-ACHA-NH_2_ in a water solution of 4 mM HClO_4_ with I = 0.1 M NaClO_4_, T = 298 K.

	Ac-AHCA-NH_2_				Ac-ACHA-NH_2_	
Species	logβ	pK_a_		logβ	pK_a_	
HL	8.72 (1)	8.72	(C)	8.76 (1)	8.76	(C)
H_2_L	15.14 (2)	6.42	(H)	15.18 (2)	6.42	(H)
Zn(II)-complexes						
ZnL	6.22 (1)	7.84	(H_2_O)	6.43 (1)	8.05	(H_2_O)
ZnH_−1_L	−1.62 (1)	9.98	(H_2_O)	−1.62 (1)	9.82	(H_2_O)
ZnH_−2_L	−11.60 (2)			−11.44 (2)		
Ni(II)-complexes						
NiL	3.86 (1)	7.39	(N^−^)			
NiH_−1_L	−3.53 (2)	8.05	(N^−^)	−3.41 (5)	8.45	(N^−^)
NiH_−2_L	−11.58 (2)	8.56	(H_2_O)	−11.86 (2)	9.63	(H_2_O)
NiH_−3_L	−20.14 (1)			−21.49 (3)		
NiL_2_				9.85 (4)		
Fe(II)-complexes						
FeL	3.71 (3)			3.55 (6)		
FeH_−2_L	−13.87 (2)	10.16	(H_2_O)	−14.18 (2)	9.42	(H_2_O)
FeH_−3_L	−23.94 (2)			−23.60 (3)		

## Data Availability

Not applicable.

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
