# Peer review of "CH vs. HC—Promiscuous Metal Sponges in Antimicrobial Peptides and Metallophores"

_molecules, 2023, doi:10.3390/molecules28103985_

Round 1
Reviewer 1 Report
The authors reported the findings on the binding patterns of Ni(II), Zn(II) and Fe(II) with Ac-ACHA-NH2 and Ac-AHCA-NH2 to evaluate the impact of the position of histidine and cysteine residues towards the coordination properties of metal(II) ions. This manuscript shares important insights into the coordination properties of metal(II) in peptides containing histidine and cysteine residues at varied positions and will be of interest to readers from related research areas. There are however comments to be addressed as below:
Abstract:
Line 25 - "biding" or "binding"
Introduction:
Line 44 - "but also" change to "and also"
Line 88 - correct the in-text citation - [18, 25, 26]
Line 105 - remove the additional space between "therefore" and "is"
Experimental
Line 168 - remove the additional space between "Fe(II)" and "to"
Results and discussion
Line 206 - suggest changing "LH2" to "H2L" as in Table 1.
Section 3.2 and onwards
Suggest standardising the reporting format of "Table 1", "Figure 1" and etc. Further, recommended using "Figure x" in the main text and "Figure Sx" in the supplementary material, to eliminate confusion in referring to the diagrams/figures discussed in the manuscript. For example, Line 210 stated Fig. S1A but the caption in the ESI is Figure 1. For Line 219, Fig. 1B was written instead. Please check and rectify throughout the manuscript.
Line 248 - remove the additional space between "," and "ZnL"
Line 260 - "consisted" or "consistent"
Line 268 - remove "." before "Figure 4B).
Author Response
Dear Editor, dear Reviewers,
Thank you very much for a detailed revision of our work. Below, we list the answers to your questions:
Reviewer 1
The authors reported the findings on the binding patterns of Ni(II), Zn(II) and Fe(II) with Ac-ACHA-NH2 and Ac-AHCA-NH2 to evaluate the impact of the position of histidine and cysteine residues towards the coordination properties of metal(II) ions. This manuscript shares important insights into the coordination properties of metal(II) in peptides containing histidine and cysteine residues at varied positions and will be of interest to readers from related research areas. There are however comments to be addressed as below:
Abstract:
Line 25 - "biding" or "binding"
Introduction:
Line 44 - "but also" change to "and also"
Line 88 - correct the in-text citation - [18, 25, 26]
Line 105 - remove the additional space between "therefore" and "is"
Experimental
Line 168 - remove the additional space between "Fe(II)" and "to"
Results and discussion
Line 206 - suggest changing "LH2" to "H2L" as in Table 1.
Section 3.2 and onwards
Suggest standardising the reporting format of "Table 1", "Figure 1" and etc. Further, recommended using "Figure x" in the main text and "Figure Sx" in the supplementary material, to eliminate confusion in referring to the diagrams/figures discussed in the manuscript. For example, Line 210 stated Fig. S1A but the caption in the ESI is Figure 1. For Line 219, Fig. 1B was written instead. Please check and rectify throughout the manuscript.
Line 248 - remove the additional space between "," and "ZnL"
Line 260 - "consisted" or "consistent"
Line 268 - remove "." before "Figure 4B).
We have made the appropriate corrections, thank you very much for the careful reading of the manuscript – it is now really improved.
Reviewer 2 Report
The work of Rowinska-Zyrek and co-workers focuses on the important topic of biomolecular self-assembly of peptide analogues with divalent metal ions of Fe, ni and zn. Knowledge gained from such studies can be transferred too the next generation drugs and/or overall better understanding of the homeostasis of living organisms. Herein studied Cys-His and His-Cys motifs and the stated hypotheses as well as proposed answers for them are of sufficient novelty and interest for this journal. Experiments were conducted with appropriate care and described so that reproducibility should be not a problem. The following can be done to strengthen the results of the paper:
A small paragraph in the introduction could be dedicated to the biomimetic artificial ligands with SH/NH groups and their metal complexes. The effect on bioactivity can be profound (e.g. DOI: 10.3390/molecules24173173) but also on other properties like e.g. catalysis (see DOI: 10.1016/j.apcata.2020.117665 or DOI: 10.1039/d2cc04015h)
Can the authors support the key results of the paper with NMR spectroscopy? I imagine for chosen pH values one could perform those for Zn ions, maybe for Ni and Fe as well, given their complexes are diamagnetic (hard to say a priori).
English is ok.
Author Response
Dear Editor, dear Reviewers,
Thank you very much for a detailed revision of our work. Below, we list the answers to your questions:
Reviewer 2
The work of Rowinska-Zyrek and co-workers focuses on the important topic of biomolecular self-assembly of peptide analogues with divalent metal ions of Fe, ni and zn. Knowledge gained from such studies can be transferred too the next generation drugs and/or overall better understanding of the homeostasis of living organisms. Herein studied Cys-His and His-Cys motifs and the stated hypotheses as well as proposed answers for them are of sufficient novelty and interest for this journal. Experiments were conducted with appropriate care and described so that reproducibility should be not a problem. The following can be done to strengthen the results of the paper:
A small paragraph in the introduction could be dedicated to the biomimetic artificial ligands with SH/NH groups and their metal complexes. The effect on bioactivity can be profound (e.g. DOI: 10.3390/molecules24173173) but also on other properties like e.g. catalysis (see DOI: 10.1016/j.apcata.2020.117665 or DOI: 10.1039/d2cc04015h)
Thank you for this suggestion. We have now included a short paragraph about this matter: “Knowledge about the coordination mode, structure and function of specific binding sites in biomimetic artificial ligands with SH/NH groups and their metal complexes biomolecules provides us knowledge that can be transferred to a new generation of materials and applications, such as catalysis [https://pubs.rsc.org/en/content/articlelanding/2022/cc/d2cc04015h], bioassays [https://pubmed.ncbi.nlm.nih.gov/31480486/] or general peptide-governed metal sorption [https://www.mdpi.com/2297-8739/9/11/370, https://www.sciencedirect.com/science/article/pii/S157002322100180X]. It is important to note that often very seemingly similar coordination motifs (as also in the case of this work) do not necessarily lead to the same coordinational and functional outcome, which is an important factor to consider in the design of novel biomimetics, catalysts and materials [https://www.sciencedirect.com/science/article/abs/pii/S0926860X20302581].”
Can the authors support the key results of the paper with NMR spectroscopy? I imagine for chosen pH values one could perform those for Zn ions, maybe for Ni and Fe as well, given their complexes are diamagnetic (hard to say a priori).
This issue is definitely worth considering. We may guess that Ni(II) (and maybe also Fe(II)) complexes will be paramagnetic at around physiological pH, later turning into diamagnetic complexes after the Ni(II) complex becomes square planar. Also the binding site of the paramagnetic complexes could be studied by NMR, provided that very small metal aliquotes are added. However, in the 5 day timeframe that we were given to revise this paper, it is not possible to carry out all NMR experiments for all studied ligands and complexes.
Once again, thank you very much for the input - it helped us to improve the manuscript.
Reviewer 3 Report
The paper of Magdalena Rowińska-Żyrek and co-authors is a fundamental work on synthesis Zn(II), Ni(II) and Fe(II) coordination compounds with model peptides in purpose of studying the order of amino acid residues to bind metal ions. The introduction contains comprehensive information about the ions of the studied metals and some information about their binding to amino acids. I have a few clarifying questions for the manuscript.
In my humble opinion, the title of the article does not match its content. For example, the phrase "metal sponge" is used, which are usually used to describe sorption processes. In the research of the authors, sorption is not studied, but only the binding of organic ligands to the metal center is estimated. It may be worth supplementing the introduction with information on the sorption of metals by such peptides. It can be seen from the work of the authors that with the simultaneous content of zinc, iron and nickel ions, selective sorption is possible with respect to nickel - this information, this conclusion would “decorate” the work.
Some answers to the questions asked by the authors in the last paragraph of the introduction are obvious. For the most part, the research done by the authors is in the field of the theory of Hard and Soft Acids and Bases (Lewis + Pearson). The authors concluded that “ii) His-Cys and Cys-His ligands do have a strong metal preference” - but isn't it obvious that differently constructed ligands react differently? I'm sorry, but maybe I don't understand the deep idea of the work. I would also add to reference [2] the original source of Pearson theory https://pubs.acs.org/doi/abs/10.1021/ja00905a001.
For better perception, add please a scheme of the structure of model peptides Ac-AHCA-NH2 and Ac-ACHA-NH2
The choice of metal anions - chlorates - is not clear. Why not choose chlorides or bromides?
I have a number of questions regarding the synthesis and characterization of compounds: please add the classic experimental part to the article (with the masses of initial samples, product yields). Is it possible to isolate the resulting complexes in an individual state and make MALDI tof? Or at least get the IR spectra of the compounds? UV-Vis spectra are usually accompanied by quantum chemical calculations to confirm transition - have authors done any calculations to confirm the d-d transition?
Line 247: “The Zn(II) coordination to Ac-ACHA-NH2” - usually, after all, the ligand is coordinated to the metal, and not the metal to the ligand. In organometallic chemistry, electron back donation is possible, but not in this case.
Line 286: “UV-Vis spectra at around 430 nm. (Figure 4B)” - need to clarify that this image is in additional materials
The authors also use the expressions "free zinc/iron/nickel" - what is included in this concept? In what medium were the curves taken? Most likely, the authors meant to say "chlorates without ligand"?
Despite the questions I have asked, I recommend accepting this article for publication in Molecules so that other researchers can start a discussion.
